# Physiological ER stress caused by amylase production induces regulated Ire1-dependent mRNA decay in *Aspergillus oryzae*

Mizuki Tanaka [1✉], Silai Zhang[2], Shun Sato[2], Jun-ichi Yokota[2], Yuko Sugiyama[2], Yasuaki Kawarasaki[3], Youhei Yamagata[1], Katsuya Gomi [2,4✉] & Takahiro Shintani [2✉]

Regulated Ire1-dependent decay (RIDD) is a feedback mechanism in which the endoribonuclease Ire1 cleaves endoplasmic reticulum (ER)-localized mRNAs encoding secretory and membrane proteins in eukaryotic cells under ER stress. RIDD is artificially induced by chemicals that generate ER stress; however, its importance under physiological conditions remains unclear. Here, we demonstrate the occurrence of RIDD in filamentous fungus using *Aspergillus oryzae* as a model, which secretes copious amounts of amylases. α-Amylase mRNA was rapidly degraded by IreA, an Ire1 ortholog, depending on its ER-associated translation when mycelia were treated with dithiothreitol, an ER-stress inducer. The mRNA encoding maltose permease MalP, a prerequisite for the induction of amylolytic genes, was also identified as an RIDD target. Importantly, RIDD of *malP* mRNA is triggered by inducing amylase production without any artificial ER stress inducer. Our data provide the evidence that RIDD occurs in eukaryotic microorganisms under physiological ER stress.

[1] Department of Applied Biological Chemistry, Graduate School of Agriculture, Tokyo University of Agriculture and Technology, Tokyo 183-8509, Japan. [2] Department of Agricultural Chemistry, Graduate School of Agricultural Science, Tohoku University, Sendai 980-8572, Japan. [3] Biomolecular Engineering Laboratory, School of Food and Nutritional Science, University of Shizuoka, Shizuoka 422-8526, Japan. [4] Laboratory of Fermentation Microbiology, Department of Agricultural Chemistry, Graduate School of Agricultural Science, Tohoku University, Sendai 980-8572, Japan. ✉email: mizuki-tanaka@go.tuat.ac.jp; katsuya.gomi.a6@tohoku.ac.jp; takahiro.shintani.d7@tohoku.ac.jp

Eukaryotic mRNAs encoding secretory and membrane proteins (SMPs) are translated by ribosomes on the endoplasmic reticulum (ER). Misfolded SMPs accumulate in the ER and induce an unfolded protein response (UPR) to counteract ER stress by increasing the capacity for protein folding and degradation of misfolded SMPs[1–3]. In the UPR, the ER-localized transmembrane kinase/endoribonuclease Ire1 is activated to splice an intron of the pre-mRNA encoding Hac1/XBP1, a basic leucine zipper-type transcription factor. The functional transcription factor translated from this unconventionally spliced mRNA induces the expression of ER chaperone and ER-associated degradation (ERAD) genes to refold and degrade unfolded SMPs, respectively. The UPR pathway is well-conserved from yeast to humans, except for the fission yeast *Schizosaccharomyces pombe*[4].

In addition to splicing Hac1/XBP1 mRNA, metazoan Ire1 cleaves ER-localized mRNAs to reduce the load of nascent polypeptides entering the ER[5]. Two mRNA fragments generated by Ire1 cleavage are rapidly degraded: the 5′-mRNA fragment without the poly(A) tail is degraded by the 3′–5′ exonuclease complex exosome in cooperation with the Ski2–Ski3–Ski8 (Ski) complex, and the 3′-mRNA fragment is degraded by the 5′–3′ exoribonuclease Xrn1. This novel mRNA degradation pathway is called regulated Ire1-dependent mRNA decay (RIDD). In 2006, RIDD was discovered in the fruit fly *Drosophila melanogaster*[6] and is now widely found in mammalian cells, plants, *S. pombe*, and the human pathogenic yeast *Candida glabrata*[7–11] but not in the budding yeast *Saccharomyces cerevisiae*.

Filamentous fungi produce large amounts of secretory hydrolytic enzymes; thus, many fungi are industrially used as enzyme sources[12,13]. *Aspergillus oryzae* is used for the production of sake, soy sauce, and soybean paste (miso) in Japan, and it secretes copious amounts of amylolytic and proteolytic enzymes required for the efficient utilization of starch and proteins, respectively, present in raw materials[14–16]. In *Aspergillus* fungi, the expression of amylolytic genes is induced by the transcriptional regulator AmyR in the presence of maltose[15,16]. In *Aspergillus nidulans*, isomaltose converted from maltose by the transglycosylation activity of secreted α-glucosidase induces AmyR activation[17]. However, the mechanism of AmyR activation in *A. oryzae* is more complex: maltose is incorporated by the maltose permease MalP and converted to isomaltose by the transglycosylation activity of the intracellular α-glucosidase MalT[18]. This unique conversion mechanism from maltose to isomaltose probably contributes to the high capacity of amylolytic enzyme production in *A. oryzae*. When *A. oryzae* produces amylolytic enzyme, UPR is induced and IreA (an ortholog of yeast Ire1) is essential for mycelial growth[19].

The UPR in filamentous fungi, including *A. oryzae*, has been well studied under forced ER stress induced by the addition of chemicals such as dithiothreitol (DTT) and tunicamycin, as well as by forced high-level expression of secretory homologous or heterologous proteins[20–24]. In addition to UPR induction, the abundance of mRNAs encoding secretory proteins decreases when ER stress is induced by the addition of DTT in filamentous fungi[24–28]. The DTT-dependent decrease in the mRNA levels of genes encoding secretory hydrolases was proposed to be due to transcriptional repression and was termed repression under secretion stress (RESS)[25,26]. In addition, ER stress reduces the transcript level of *amyR* in *Aspergillus niger* and *A. oryzae*[22,29,30]. Therefore, in filamentous fungi, the decrease in mRNA levels induced by ER stress is believed to be due to transcriptional repression; however, reports on RIDD are lacking.

In this study, we demonstrated the presence of RIDD in filamentous fungi. Using a modified α-amylase gene with a mutation in the secretory signal sequence, we found that the DTT-induced decrease in the α-amylase mRNA level was dependent on its ER-associated translation and IreA in *A. oryzae*. In addition, we

found that physiological ER stress induced by amylolytic enzyme production strongly induces RIDD of *malP* mRNA. Our data highlight the potential for *A. oryzae*, in which both UPR and RIDD are induced by physiological ER stress, to be a useful model for elucidating the physiological significance of the ER stress response in eukaryotic microorganisms.

## Results

**Rapid decrease in α-amylase mRNA level after adding DTT.** Studies have shown that ER stress elicited with chemicals such as DTT rapidly downregulates the expression of the major secreted hydrolytic enzymes in filamentous fungi, including *A. oryzae*[28,30]. To examine the selective repression of transcription of secretory protein-coding genes upon the addition of DTT in more detail, we monitored the mRNA levels of genes encoding secretory α-amylase (*amyA*, *amyB*, and *amyC*) and glucoamylase (*glaA*), together with the genes encoding the ER chaperone, BipA (*bipA*), and the cytoplasmic enolase (*enoA*) for a shorter period after DTT was added to the culture of the *A. oryzae* RIB40 strain grown in YPM medium (Fig. 1, left panels). *A. oryzae* RIB40 contains three copies of α-amylase genes, *amyA*, *amyB*, and *amyC*, the sequences of which are almost identical[31]. The levels of the *amyA/B/C* and *glaA* mRNAs decreased dramatically within 30 min after the addition of DTT. As expected, the expression level of *bipA* increased markedly upon adding DTT, confirming that the addition of DTT induced the UPR. In contrast, the *enoA* mRNA level was unaffected by DTT addition. DTT-induced downregulation of secreted protein expression is believed to result from transcriptional repression, which depends on the *cis*-elements in the promoter regions of their genes and decreased expression of *trans*-acting activators[21,25,26]. Accordingly, we substituted the promoter sequence of *amyB* with that of actin (*actA*), the expression of which was not affected by DTT treatment (Supplementary Fig. 1). The *actAp-amyB* DNA construct was introduced into an *A. oryzae* strain lacking all three copies of the intrinsic α-amylase genes (Δ*amyA/B/C*), and the *amyB* mRNA level was examined. Importantly, we found that the *amyB* mRNA level decreased, similar to that expressed from the *amyB* promoter (Fig. 1). These results suggested that a mechanism other than transcriptional repression was involved in the rapid decrease in the *amyA/B/C* mRNA levels caused by DTT addition.

**Decrease in α-amylase mRNA level depended on the secretory signal sequence.** Previous and present studies have shown a rapid decrease in mRNA levels in the DTT-treated condition,

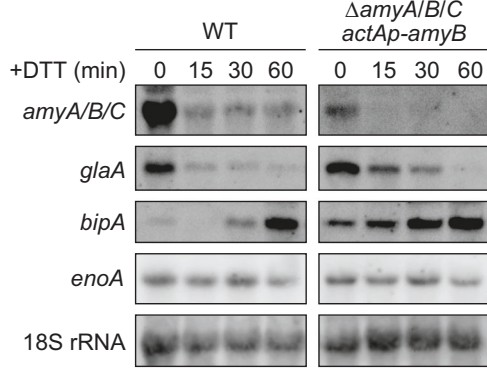

**Fig. 1 Expression analyses of α-amylase (*amyA*, *amyB*, and *amyC*) and glucoamylase A (*glaA*) genes after DTT treatment.** Total RNAs were extracted from mycelia treated with DTT and subjected to northern blot analysis with DNAs encoding *amyB*, *glaA*, *bipA*, *enoA*, and 18 S rRNA as the probes.

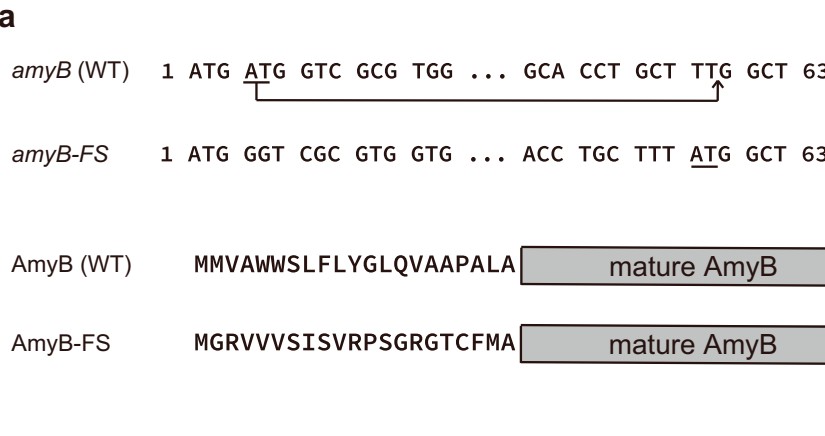

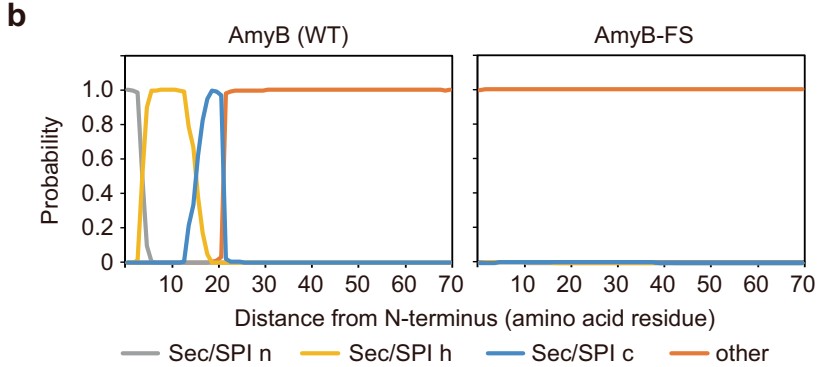

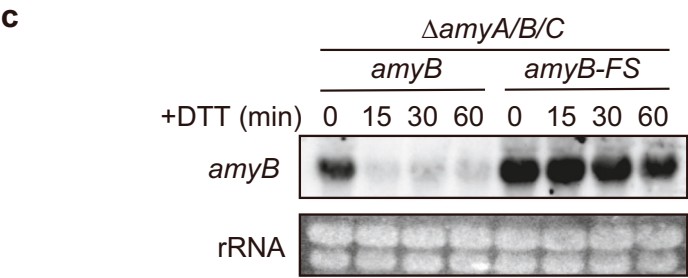

**Fig. 2 Effect of a frameshift mutation in the signal peptide of AmyB on the *amyB* mRNA level. a** The nucleotide sequences of the 5′-CDS region of the wild type (*amyB*) and the frameshift mutant (*amyB-FS*) of *amyB* and their deduced amino acid sequences. The underlines indicate the deleted and inserted nucleotides. **b** Prediction of signal peptide of the wild type and mutant AmyB using SignalP 6.0. The likelihood scores for the signal peptides were 0.9992 and 0 for wild type and mutant AmyB, respectively. Sec/SPI; secretory signal peptides transported by the Sec translocon and cleaved by signal peptidase I. The characters "n", "h", and "c" indicate the N-terminal region, the center hydrophobic region, and the C-terminal region of the signal peptide, respectively. **c** Northern blot analysis of transcripts prepared from the DTT-treated *ΔamyA/B/C* strain expressing the wild type or mutant *amyB* genes.

specifically of those encoding secretory proteins[25,26]. Therefore, we generated modified *amyB* with a mutation in the secretory signal sequence to investigate whether the reduction in the α-amylase mRNA level by DTT was dependent on ER targeting. In the modified *amyB*, termed *amyB-FS*, two nucleotides (A and T) in the second codon (ATG) were deleted and inserted near the end of the secretory signal sequence to introduce a frameshift in the signal peptide sequence of AmyB without compromising the overall mRNA structure (Fig. 2a). The *amyB-FS* gene is expected to encode a non-functional secretory signal sequence due to a frameshift mutation, as shown using SignalP-6.0 prediction (https://services.healthtech. dtu.dk/service.php?SignalP-6.0) (Fig. 2b). A plasmid harboring an intact or modified *amyB* was introduced into the chromosome of the *ΔamyA/B/C* strain. A decrease in the mRNA level of *amyB-FS* was not observed compared to that of intact *amyB*

upon DTT treatment, although the *amyB-FS* mRNA was still gradually decreased (Fig. 2c). These results suggested that the decrease in the α-amylase mRNA level was dependent on its localization in the ER.

**α-amylase mRNA was cleaved by IreA.** As the reduction in the α-amylase mRNA level upon DTT treatment was dependent on its targeting to the ER, we assumed that RIDD might be involved in this phenomenon. To test this hypothesis, we utilized a knockdown strain of *ireA* (*nmtAp-ireA*), the expression of which was repressed in the presence of thiamine under the control of the *nmtA* promoter, as *ireA* is an essential gene in *A. oryzae*[23]. Mycelia grown in the presence of thiamine were transferred to a maltose-containing medium (still containing thiamine) to induce α-amylase expression, followed by the addition of DTT to the medium. We found

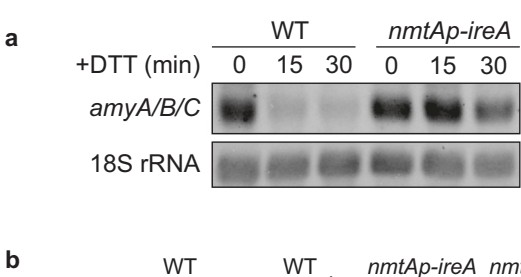

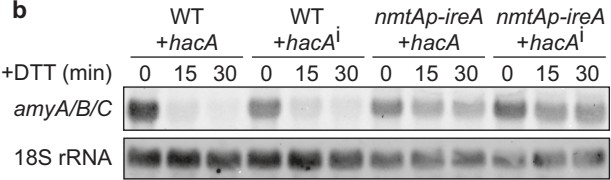

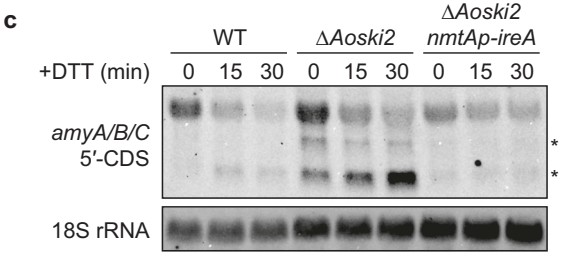

**Fig. 3 Involvement of IreA in the DTT-induced reduction of the α-amylase mRNA. a** Northern blot analysis of the *amyA/B/C* mRNAs in *ireA*-repressing strain treated with DTT. After induction of α-amylase gene expression, DTT was added to the culture medium and the mycelia were harvested at the indicated times. **b** Northern blot analysis of the *amyA/B/C* mRNAs in *hacA*-expressing strains treated with DTT. **c** Northern blot analysis of the *amyA/B/C* mRNAs in WT, Δ*Aoski2*, and Δ*Aoski2*/*nmtAp-ireA* strains treated with DTT. The asterisks indicate short mRNA fragments.

that the decrease in the *amyA/B/C* mRNA levels was markedly delayed by *ireA* repression (Fig. 3a), indicating that IreA plays an important role in reducing the *amyA/B/C* mRNAs under ER stress. We next confirmed whether suppression of the decline in the *amyA/B/C* mRNA levels was directly caused by *ireA* repression and not by the inefficient processing of the *hacA* mRNA. Toward this, we expressed artificially activated *hacA* (*hacA*i), which lacks the 20-base intron spliced from the *hacA* pre-mRNA by IreA, in the *nmtAp-ireA* strain[23] and found that the expression of *hacA*i did not affect the IreA-dependent decrease in the *amyA/B/C* mRNA levels induced by DTT treatment (Fig. 3b). This indicated that the decrease in the *amyA/B/C* mRNA levels was controlled by IreA but not by the downstream events of HacA activation. If the *amyA/B/C* mRNAs are cleaved by IreA, the resulting mRNA fragments lacking a poly(A) tail are expected to be rapidly degraded by the 3′ to 5′ mRNA degradation pathway. To test this hypothesis, we generated a disruption mutant of an ortholog of the RNA helicase Ski2 (AoSki2), which is a component of the Ski complex mediating 3′ to 5′ mRNA degradation by the cytoplasmic exosome. Northern blot analysis using the 5′-terminal region of the *amyA/B/C* coding sequence (CDS) as the probe showed that the abundance of the short fragments of the *amyA/B/C* mRNAs increased noticeably after *Aoski2* deletion, which is dependent on DTT treatment (Fig. 3c). Furthermore, the accumulation of short fragments was severely inhibited by *ireA* repression (Fig. 3c), indicating that the *amyA/B/C* mRNAs were cleaved in an IreA-dependent manner in response to the ER stress induced by DTT addition. Taken together, our observations suggested that the rapid decrease in the *amyA/B/C* mRNA levels upon DTT treatment was mainly mediated by RIDD.

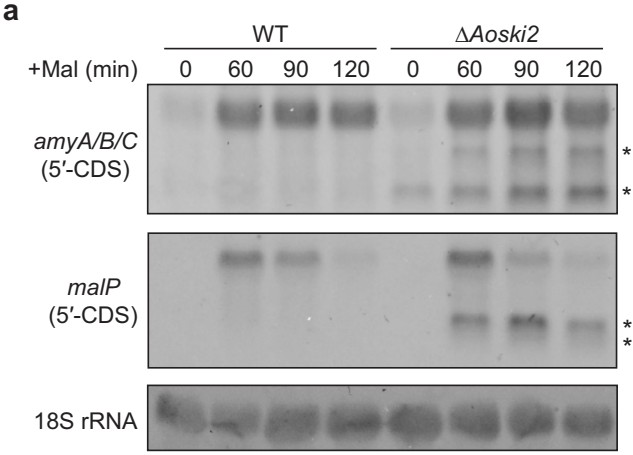

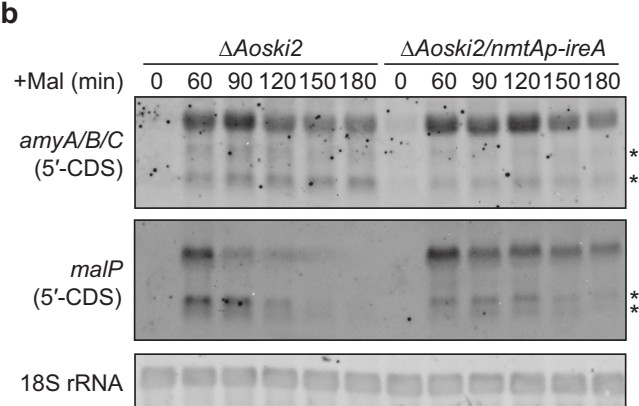

**Fig. 4 Expression analysis of *amyA/B/C* and *malP* after maltose addition without DTT treatment. a** Northern blot analysis of the *amyA/B/C* and *malP* mRNAs in wild type and Δ*Aoski2* strains. The mycelia were harvested after incubating in maltose medium for the indicated times to extract total RNAs. The asterisks indicate short mRNA fragments. **b** Northern blot analysis of the *amyA/B/C* and *malP* mRNAs in Δ*Aoski2* and Δ*Aoski2*/*nmtAp-ireA* strains. The mycelia were harvested after incubating in maltose medium containing 20 μM thiamine for the indicated times. The asterisks indicate short mRNA fragments.

**mRNAs of α-amylase and *malP* were cleaved under physiological condition**. Although DTT treatment noticeably increased the levels of the short fragments of the *amyA/B/C* mRNAs, they were detectable even before DTT treatment in the Δ*Aoski2* strain (Fig. 3c). As UPR is induced by amylolytic enzyme production in *A. oryzae*[23], we hypothesized that the production of amylolytic enzymes alone can trigger RIDD in *A. oryzae*. To test this hypothesis, we examined the expression of *amyA/B/C* in the Δ*Aoski2* strain after incubation in liquid maltose medium without adding DTT. Short fragments of the *amyA/B/C* mRNAs were consistently observed along with the full length *amyA/B/C* mRNAs after adding maltose (Fig. 4a). When *ireA* expression was repressed in Δ*Aoski2* strain, accumulation of the short fragments of the *amyA/B/C* mRNAs was reduced, whereas the expression level of the full length *amyA/B/C* mRNAs was not affected (Fig. 4b, Supplementary Fig. 2). This suggested that IreA-dependent cleavage of the *amyA/B/C* mRNAs occurred even without DTT. Interestingly, we found that the *malP* mRNA, a prerequisite for the induction of amylolytic genes by maltose in *A. oryzae*, was degraded in an IreA-dependent manner. The short fragment of the *malP* mRNA was observed in the Δ*Aoski2* strain when the 5′-CDS of *malP* was used

as a probe (Fig. 4a). The level of the full length *malP* mRNA was dramatically reduced between 60 and 90 min after maltose addition, whereas those of the shorter fragments were not. These results suggested that the full length *malP* mRNA was cleaved for degradation, and that the resulting shorter *malP* mRNA might transiently emerge and be degraded by cytoplasmic exosomes. When *ireA* expression was repressed in the Δ*Aoski2* strain, the decrease in the level of the full length *malP* mRNA was delayed, accompanied by the delayed appearance of the shorter mRNA (Fig. 4b), suggesting that the *malP* mRNA might be cleaved by IreA under the condition of amylolytic enzyme production. To confirm whether the *malP* mRNA cleavage correlated with ER stress, the *malP* mRNA was detected after DTT treatment of the Δ*Aoski2* strain. The short *malP* mRNA fragments were rapidly generated along with the short *amyA/B/C* mRNA fragments in the Δ*Aoski2* strain treated with DTT (Supplementary Fig. 3a), which was repressed by *ireA* repression (Supplementary Fig. 3b). Taken together, these results indicated that the *malP* mRNA is degraded in an IreA-dependent manner under ER stress.

**RIDD of *malP* mRNA was induced by the production of amylolytic enzymes.** We have previously shown that UPR induction by maltose can be avoided by deletion of the transcription factor AmyR, which induces the expression of amylolytic genes upon activation by maltose[23]. Therefore, we decided to verify whether *malP* mRNA cleavage, observed when cultured in maltose medium, could be induced by the production of amylolytic enzymes. Toward this, we examined the *malP* transcripts in the Δ*Aoski2*Δ*amyR* double-disruption mutant. Northern blot analysis showed that the deletion of *amyR* repressed the appearance of the 5′-*malP* mRNA fragments, accompanied by a delayed decrease in the level of the full length *malP* mRNA in the Δ*Aoski2* strain (Fig. 5a). This suggested that physiological ER stress induced by amylolytic enzyme production triggers the RIDD of the *malP* mRNA. As AmyR regulates the expression of secretory α-glucosidases, which hydrolyze maltose to glucose, *amyR* disruption may affect *malP* transcription, which is repressed via carbon catabolite repression (CCR). To verify the effect of *amyR* disruption on the CCR, the *malT* mRNA encoding a cytoplasmic α-glucosidase, transcription of which is regulated cooperatively with *malP*[32,33], was detected using the 5′-CDS of *malT* as a probe. Short fragments were not observed when the *malT* mRNA was analyzed in the Δ*Aoski2* strain (Fig. 5a), confirming that mRNAs encoding cytoplasmic proteins do not become targets of RIDD. Nevertheless, *malT* transcription decreased gradually over time and *amyR* disruption suppressed this decrease. This suggested that *malT* and *malP* transcription is repressed by CCR, which is triggered by glucose generated from maltose. Importantly, however, the level of the full length *malP* transcript decreased faster than that of *malT* in the Δ*Aoski2* strain. This suggested that after maltose drastically elevates the production of amylolytic enzymes, the number of *malP* transcripts is downregulated by two different mechanisms: RIDD and CCR.

In fission yeast, *ski2* deletion leads to increased susceptibility to tunicamycin, an antibiotic widely used to induce ER stress[9]. Interestingly, the growth of the *Aoski2* disruptant was severely impaired on agar media containing maltose but not glucose (Fig. 5b). This carbon source-dependent growth defect was also observed in a disrupted strain of the Ski3 ortholog (AoSki3), another subunit of the Ski complex. *amyR* deletion rescued the growth defect of the Δ*Aoski2* and Δ*Aoski3* strains on maltose medium (Fig. 5b). This suggested that RIDD contributes to maintenance of cellular homeostasis in *A. oryzae* under conditions that produce amylolytic enzymes, although detailed information

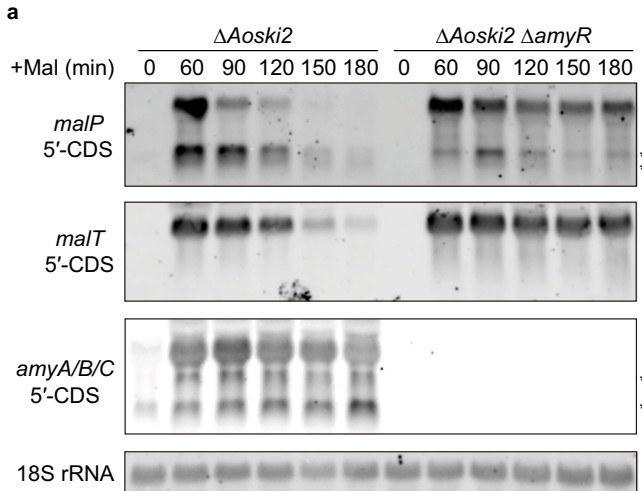

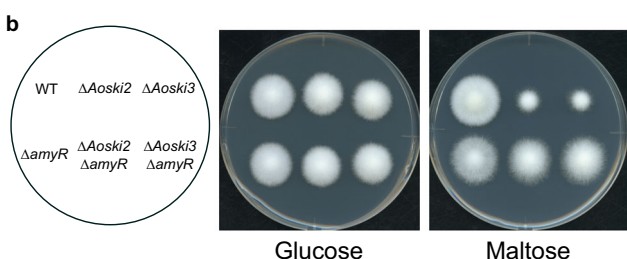

**Fig. 5 Effects of *amyR* deletion on *malP* mRNA cleavage and growth of Ski complex-deficient mutants. a** Northern blot analysis of the *amyA/B/C* and *malP* mRNAs in Δ*Aoski2* and Δ*Aoski2*Δ*amyR* strains. Total RNAs were extracted from mycelia after incubating in maltose medium for the indicated time points. The asterisks indicate short mRNA fragments. **b** Growth of single and double disruption mutant strains of the Ski complex and *amyR*. Approximately $1 \times 10^4$ conidiospores of each strain were grown for 3 days at 30 °C on minimal agar media containing 1% glucose or maltose as a sole carbon source.

regarding the roles of the 3′–5′ mRNA decay in the production of amylolytic enzymes is not available.

## Discussion
RIDD is a feedback mechanism that alleviates ER stress caused by the accumulation of misfolded proteins and/or the overload of secretory and membrane proteins in the ER. Treatment of cells with chemicals such as DTT and tunicamycin mimics the accumulation of misfolded proteins in the ER to induce RIDD in many eukaryotes. The decrease in the mRNA levels of secretory hydrolase-coding genes upon treatment with DTT is also observed in filamentous fungi, although it has been attributed to the repression of transcription, called RESS, of target genes. In this study, we showed that RIDD contributes to a DTT-induced decrease in the transcripts of secretory proteins in filamentous fungi. In addition, RIDD of the *amyA/B/C* and *malP* mRNAs occurred when amylolytic enzyme production was induced without chemical treatment. A schematic diagram showing the response to physiological ER stress induced by amylolytic enzyme production is illustrated in Fig. 6. To the best of our knowledge, this is the first study to demonstrate that RIDD occurs in eukaryotic microorganisms under natural physiological conditions.

According to previous studies and our current study, filamentous fungi appear to utilize two mechanisms, RESS and RIDD, to reduce the excess load of secretory proteins in the stressed ER. However, in our experimental condition involving DTT treatment,

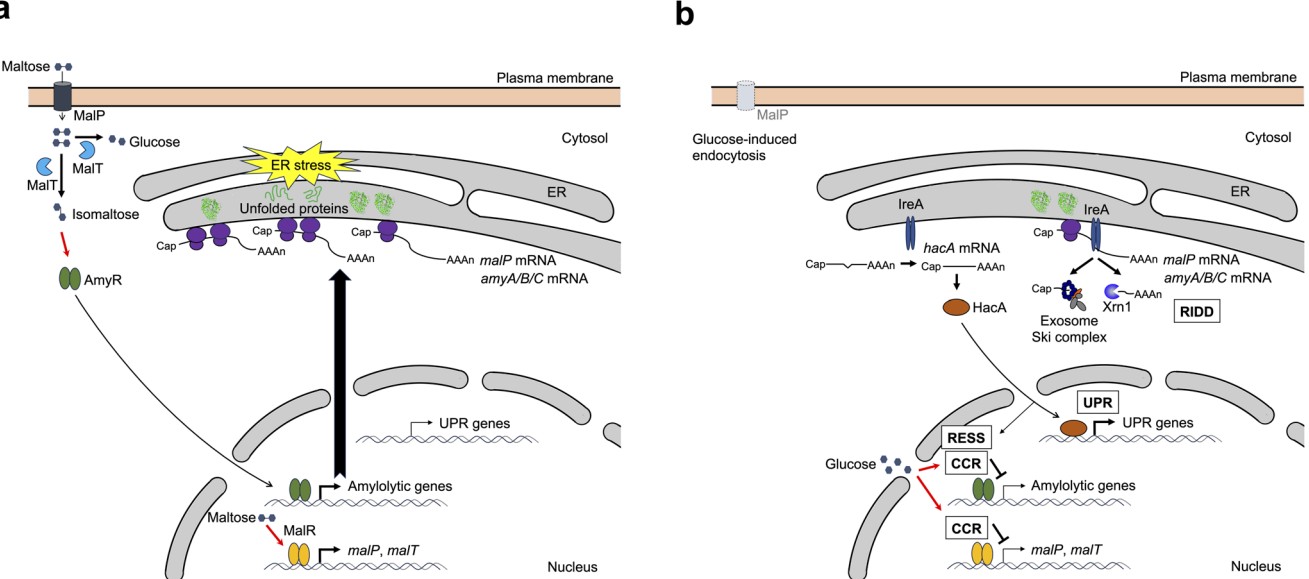

**Fig. 6 Schematic diagram showing the response to physiological ER stress induced by amylolytic enzyme production. a** Expression of amylolytic genes is induced by maltose uptake through MalP, and the unfolded proteins resulting from the synthesis of nascent polypeptides from their mRNAs induce ER stress. **b** The resulting ER stress activates IreA, leading to HacA activation and RIDD. Thereafter, RIDD and the HacA-mediated UPR and RESS alleviate ER stress. In addition to RIDD and RESS, carbon catabolite repression (CCR) and MalP endocytosis induced by glucose generated from the maltose hydrolysis also contribute to the decrease in *amyA/B/C* and *malP* mRNAs. The correctly folded protein is represented by the three-dimensional structural image of α-amylase (PDB ID: 6TAA). The red arrows indicate the activation of transcription factors and CCR.

RIDD might be the major route utilized to decrease the transcripts of secretory proteins in *A. oryzae*. This was substantiated by several lines of evidence: (i) the replacement of the transcriptional promoter of *amyB* with that of *actA* did not suppress the reduction in the *amyB* mRNA level; (ii) the frameshift mutation in the ER targeting signal of AmyB markedly suppressed the rapid decrease in its transcript; (iii) IreA, but not HacA, was required to decrease *amyA/B/C* mRNAs; (iv) degradation products of *amyA/B/C* mRNAs appeared in the *ΔAoski2* strain. Although our findings indicated that RIDD plays a major role in the reduction of transcripts encoding secretory proteins in *A. oryzae*, the slower decrease in the *amyB-FS* mRNA levels upon DTT treatment reflects the RESS of the *amyB* mRNA. In other words, the decrease in *amyB* mRNA could depend completely on RESS unless *amyB* mRNA is not targeted to the ER. Genome-wide transcriptomic analyses revealed that the transcription of *amyR* was markedly down-regulated in *hacA*[i]-expressing *A. niger* and *A. oryza*[29,30]. These observations implied that RESS is regulated by the UPR if the transcription of RESS targets is mediated by AmyR. As *ireA*-repressed cells expressing *hacA*[i] suppressed the reduction in the *amyA/B/C* mRNA levels treated with DTT, we speculated that RIDD and RESS might be regulated by different pathways. We propose that filamentous fungi use a dual mechanism for repressing the abundance of ER-localized mRNAs to alleviate ER stress. RIDD rapidly removes mRNAs localized in the ER, while RESS prevents de novo synthesis of specific mRNAs. Filamentous fungi can produce 100 g/l of secreted proteins[34] and thus may have a unique dual ER stress response system to deal with ER stress induced by the production of high levels of secretory proteins.

Maltose incorporated via MalP is transglycosylated to iso-maltose by the cytoplasmic α-glucosidase MalT to activate AmyR, thereby inducing amylolytic enzymes in *A. oryzae*[18,33,35]. We have previously shown that the addition of glucose to the media represses the expression of *malT* and *malP*[32] and accelerates the degradation of MalP via endocytosis[35]. These findings suggested that MalP-mediated maltose uptake is an upstream event in the regulation of amylolytic gene expression. Interestingly, in

addition to catabolite repression for optimizing starch metabolism, RIDD targets the *malP* transcript for overcoming the stress caused by protein overload in the ER. This observation is consistent with that of our previous report showing that the induction of amylolytic enzyme production by maltose induces the UPR[19]. These results suggested that the RIDD of the *malP* transcript might function as part of a feedback control to avoid further loading of amylolytic enzymes into the ER. Loss of the Ski complex, which accumulates mRNA remnants from the cleavage by IreA, caused severe growth inhibition in the maltose medium. Although the mechanism of toxicity remains elusive, clearance of mRNA remnants may be important for the maintenance of cellular homeostasis in *A. oryzae*.

In the present study, we identified the *amyA/B/C* and *malP* transcripts as targets of RIDD under physiological conditions. However, the efficiency of cleavage by IreA may differ; *malP* mRNA may be more susceptible to cleavage than *amyA/B/C* mRNAs. Although the underlying reason is not known, this might be attributed to the specificity of IreA for the sequence and the secondary structure of the RNAs. Mammalian Ire1a specifically cleaves the CUGCAG sequence with a stem-loop structure[36]. In filamentous fungi, an intron within *HAC1/hacA* pre-mRNA is cleaved by Ire1/IreA in the C(U/C)GCAG sequence of the stem-loop structure[37]. This suggests that the CUGCAG sequence is also a target of cleavage by IreA in filamentous fungi. A single CUGCAG sequence is present within the 5′-region of both *amyA/B/C* and *malP* mRNAs. In particular, the CUGCAG sequence in *malP* mRNA was located in the loop region of the predicted large stem-loop structure (Supplementary Fig. 4). In contrast, the appearance of multiple short fragments of both *amyA/B/C* and *malP* mRNAs suggested that these mRNAs have more than one cleavage site. Further experiments are required to determine the cleavage sites of these mRNAs. More examples of RIDD targets will also help us understand the structural features underscoring the cleavage specificity of IreA in filamentous fungi. We found that the growth of the Ski complex-deficient strains was inhibited not only by maltose and starch as carbon sources but also by xylose and xylan, inducers

of xylanolytic and cellulolytic gene expression[38] (Supplementary Fig. 5), suggesting that *A. oryzae* may harbor more targets of physiological RIDD. The identification of a wide range of RIDD targets and their cleavage sites will provide a comprehensive understanding regarding the mechanisms and roles of RIDD in filamentous fungi. The cleavage sites of the RIDD targets in most eukaryotes remain unclear[39]. Information obtained from studies on filamentous fungi will also be helpful for understanding RIDD in eukaryotic cells.

RIDD function has been well studied in metazoans[40,41]. In particular, RIDD is involved in insulin secretion via the degradation of insulin mRNA under high-glucose conditions[42,43], indicating that RIDD plays an important role under physiological conditions. In microorganisms, the molecular mechanism and role of the UPR have been well studied in *S. cerevisiae*, whereas RIDD was not found in this yeast. In contrast, recent studies revealed the molecular mechanism of RIDD in *S. pombe*, although this yeast lacks an *HAC1* ortholog. To the best of our knowledge, this is the first study to show that RIDD occurs under physiological conditions in eukaryotic microorganisms. In addition, this study underscores the potential for *A. oryzae* to be a useful model for elucidating the physiological significance of the ER stress response in eukaryotic microorganisms since both UPR and RIDD are actuated in this filamentous fungus under physiological ER stress condition. Further analysis of RIDD in *A. oryzae* will improve our understanding of its functions in microorganisms and other eukaryotes.

## Methods

**Strains and media**. *A. oryzae* NS4 (*niaD⁻*, *sC⁻*)[44] and the *ΔligD::loxP pyrG⁻* strain (*ΔligD::loxP*, *sC⁻*, *niaD⁻*, *pyrG⁻*)[45], derived from the wild type strain, *A. oryzae* RIB40 (National Research Institute of Brewing Stock Culture), were used as recipient strains for gene disruption. The *A. oryzae* strains used in this study are listed in Supplementary Table 1. *Escherichia coli* DH5α was used for the construction and propagation of plasmid DNAs. *A. oryzae* was grown in YPM medium (1% yeast extract, 1% Bacto Peptone, and 2% maltose) or Czapek–Dox (CD) medium (0.5% $(NH_4)_2SO_4$, 0.05% KCl, 0.2% $KH_2PO_4$, 0.05% $MgSO_4$, trace amounts of $FeSO_4$, $ZnSO_4$, $CuSO_4$, $MnSO_4$, $Na_2B_4O_7$, and $(NH_4)_6Mo_7O_{24}$, and 1% sugar) supplemented with 0.003% methionine, if required. Glycerol and maltose (1%) were used as carbon sources to not induce and induce amylolytic gene expression, respectively. HIPOLYPEPTON (0.1%; Nihon Pharmaceutical Co., Ltd., Tokyo, Japan) was added to promote growth when *A. oryzae* strains were pre-cultured in CD medium. For the DTT treatment, 20 mM DTT was added directly to the cultures.

**Construction of plasmids and strains**. Plasmids encoding wild type and mutant *amyB* were constructed as follows: The *A. oryzae* *amyB* was amplified using polymerase chain reaction (PCR) with oligonucleotides, oTAKA381 and oTAKA382, and then inserted into *Hin*dIII/*Xba*I-digested pUC-niaD[46] using the In-Fusion Snap Assembly master mix (Takara Bio, Shiga, Japan) to generate pUC-niaD-amyB. *amyB-FS* was obtained using QuikChange site-directed mutagenesis with pUC-niaD-amyB as the template DNA and the oligonucleotides YS3 and YS4. A plasmid expressing *amyB* under the control of the *actA* promoter was constructed as follows. The vector DNA containing *niaD* and *amyB* was PCR-amplified with pUC-niaD-amyB as the template DNA and oligonucleotides, vector-F and vector-R. The resulting vector fragment was assembled with the *actA* promoter PCR-amplified with the oligonucleotides, PactA-F and PactA-R using In-Fusion Snap Assembly master mix to generate pUC-niaD-PactA-amyB.

These plasmids were linearized using *Nsi*I and transformed in *A. oryzae ΔamyA/B/C* cells.

The plasmid for *Aoski2* (AO0900110006369) disruption was constructed as follows: A DNA fragment harboring approximately 1.0 kb upstream sequence of *Aoski2* was amplified using PCR and the ski2upsenKpnI and ski2upantiBamHI primers. The amplified fragment was digested with *Kpn*I and *Bam*HI and inserted into *Kpn*I/*Bam*HI-digested pUSC[44], yielding pUSC-ski2up. A DNA fragment harboring approximately 1.0 kb downstream sequence of *Aoski2* was amplified using the ski2downsenPstI and ski2downantiPstI primers. The amplified fragment was digested with *Pst*I and inserted into *Pst*I-digested pUSC-ski2up, yielding pUSC-Δski2.

The plasmid for *Aoski3* (AO090005001388) disruption was constructed as follows: A DNA fragment harboring approximately 1.0 kb upstream sequence of the *Aoski3* 5′-region was amplified using the ski3upsenBamHI and ski3upantiXbaI primers. The amplified fragment was digested with *Bam*HI and *Xba*I and inserted into the *Bam*HI/*Xba*I-digested pUSC, yielding pUSC-ski3up. A DNA fragment harboring the *Aoski3* 3′-region was also amplified using the ski3downsenPstI and ski3downantiSphI primers. The amplified fragment was digested with *Pst*I and *Sph*I, and inserted into *Pst*I/*Sph*I-digested pUSC-ski3up, yielding pUSC-Δski3.

The plasmid for the replacement of the *ireA* upstream region with the *nmtA* promoter using the *A. nidulans* orotidine-5′-decarboxylase gene (*pyrG*) as the selectable marker was constructed as follows. The DNA fragment of the *nmtA* promoter associated with the *ireA* coding region was obtained from *Pst*I/*Sph*I-digested psCPnmtAireA[19]. The obtained DNA fragment was replaced with the *ireA* downstream region excised from *Pst*I/*Sph*I-digested pΔireA::pyrG[19] using two distinct ligation reactions, yielding pnmtAp-ireA::pyrG. The DNA fragment obtained from the *Spe*I/*Sph*I-digested pnmtAp-ireA::pyrG was used to transform the *ΔAoski2 pyrG⁻* strain.

The DNA fragment for *amyR* disruption was obtained from the plasmid pΔamyR::pyrG, as described previously[19].

The nucleotide sequences of all primers used for gene disruption are shown in Supplementary Table 2.

**Fungal transformation**. DNA fragments for gene disruption were introduced into *A. oryzae* using the polyethylene glycol (PEG)-protoplast method[47], and Yatalase (Takara Bio Inc.) was used for protoplast preparation. To complement *pyrG* deficiency, plasmid pUC/pyrG/niaD[48] was introduced into the *ΔAoski2* and *ΔAoski3* mutants using *ΔligD::loxP pyrG⁻* as the host strain.

**Fungal cultivation**. To examine the reduction in α-amylase mRNA levels after DTT addition, the WT (RIB40) and *ΔamyA/B/C* strains expressing the WT and mutant *amyB*, respectively, were grown in YPM medium at 30 °C for 24 h and treated with 20 mM DTT.

To examine the involvement of IreA in the DTT-induced reduction of α-amylase mRNA levels, the WT (*ΔligD::loxP pyrG::niaD*) and *nmtAp-ireA* strains were pre-cultured in CD (glycerol) medium containing 0.1% HIPOLYPEPTON and 20 μM thiamine at 30 °C for 24 h, and the mycelia were resuspended in CD (maltose) medium containing 20 μM thiamine. After 60 min-incubation at 30 °C, DTT was added to the medium to a final concentration of 20 mM.

To examine the *amyA/B/C* and *malP* mRNA levels after maltose addition without DTT treatment, the WT and *ΔAoski2* strains were pre-cultured in CD (glycerol) medium containing 0.1% HIPOLYPEPTON at 30 °C for 24 h, and the mycelia were resuspended in CD (maltose) medium.

**Northern blot analysis**. *A. oryzae* mycelia cultured in the liquid medium were harvested over time using an open-ended pipette

and Miracloth (Merck Millipore, Billerica, MA, USA), washed with distilled water, and then frozen in liquid nitrogen. The mycelia were ground into a fine powder in liquid nitrogen using a mortar and pestle. The powdered mycelia were suspended in ISOGEN (Nippon Gene Co., Ltd., Tokyo, Japan) and total RNA was extracted according to the manufacturer's instructions. Northern blot analysis was performed as described previously[49]. Digoxigenin (DIG)-labeled DNA fragments were synthesized using PCR with the primers shown in Supplementary Table 3 and were used as probes.

**Statistics and reproducibility**. All experiments were repeated at least twice independently as biological replications and the results are reproducible. All attempts at replication were successful.

**Reporting summary**. Further information on research design is available in the Nature Portfolio Reporting Summary linked to this article.

## Data availability

Original uncropped and unedited northern blot images for all figures are shown in Supplementary Figs. 6, 7, 8, 9, and 10, respectively. Source data for boxplot in Supplementary Fig. 2 can be found in Supplementary Data 1.

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

## Acknowledgements

We thank Dr. Osamu Mizutani for providing the Δ*ligD*::*loxP pyrG⁻* mutant strain. This study was supported by JSPS KAKENHI (grant numbers 09J06211 and 16K15084), the JSPS Core-to-Core Program (Advanced Research Networks) titled "Establishment of international agricultural immunology research-core for a quantum improvement in food safety", and an NISR Research Grant from the Noda Institute for Scientific Research, Japan. We would like to thank Editage (www.editage.com) for English language editing.

## Author contributions

M.T., K.G., and T.S. designed the study; M.T., S.S., S.Z., J-I.Y., and Y.S. performed the study; M.T., K.G., Y.K., Y.Y., and T.S. analyzed the data; M.T., K.G., and T.S. wrote the paper.

## Competing interests

The authors declare no competing interests.
