## [Peer Review File · Communications Biology]

Reviewers' comments:

Reviewer #1 (Remarks to the Author):

In the manuscript by Tanaka et al the authors investigate the mechanism underlying the phenomenon of reduced transcript levels of genes encoding secreted proteins under ER stress conditions in *Aspergillus oryzae*. Using the constitutive *actA* promoter, the authors demonstrate that reduced transcript levels of *amyA/B/C* and *glaA* after induced ER stress are not (only) based on the previously coined term RESS (repression under secretion stress). They furthermore demonstrate convincingly the requirement of ER targeting and the presence of IreA for degradation of the *amyB* and *malP* mRNA. Reduced degradation in the Δ *ski2* background further suggests that the *amyA/B/C* and *malP* mRNAs are degraded via RIDD (regulated Ire1-dependent decay) and the 3'-5' exosomal degradation pathway. Importantly, RIDD of *amyA/B/C* and of *malP* mRNAs is induced under physiological conditions. Taken together, the authors reveal that RIDD, which was previously thought to occur only in higher eukaryotes and fungi without a functional Ire1/Hac1 pathway (*S. pombe* and *Candida glabrata*), functions in *Aspergillus oryzae* (and likely also other *Aspergilli*) to reduce the folding demand in/of the ER. This paper is very well and clearly written. Intro/abstract and discussion are on point. It has not happened very often to me that I do not have any major or minor points to be addressed. Congratulations to this wonderful piece of work adding truly novel and important insight, expanding the knowledge on the unfolded protein response in fungi and in general.

Reviewer #2 (Remarks to the Author):

In this manuscript, the authors insist that Regulated Ire1-dependent decay (RIDD) occurs under physiological ER stress in *Aspergillus oryzae*. To support this, the authors treated DTT and checked mRNA levels of *amyA/B/C*. The authors also used *ireA*, *hacA*, and *ski2* mutant strains and examined *amyA/B/C* and *malP* expression. It might be useful for understand the ER stress response. There are some comments.

- The authors can detect *amyB* in Figure 2. Then, it seems possible to check the expression of the *amyA*, *amyB*, and *amyC* genes.
- It's much better to focus on the *amyB* expression.
- The *amyB* protein production or enzyme activity should be examined.
- The cleavage pattern of *amyA/B/C* in Figures 1 and 3 can be provided.
- The summary figure can be useful for understand this manuscript.

Reviewer #3 (Remarks to the Author):

In this manuscript, the authors found that the mechanism of regulatory Ire1-dependent decay (RIDD) exists in filamentous fungi, *Aspergillus oryzae*, and they also found the physiological conditions under which RIDD occurs. The authors found that RIDD of *malP* mRNA is strongly induced by the production of amyolytic enzymes, rather than by chemical treatments that induce an endoplasmic reticulum stress response. These results are very interesting because they show that, in *Aspergillus oryzae*, both UPR and RIDD are induced by physiological endoplasmic reticulum stress. Although the experiments are carefully conducted and the paper is carefully written, some parts are a bit difficult to read. The following points should be further addressed or modified.

Major points

Fig. 3: The authors should compare the *ireA* mutant strain with the *hacA* mutant strain.

Fig. 2 and Fig. 4: The frame shift mutant applied to *amyB* should be analyzed with *malP* in Fig. 4.

Minor points

- 1) In Fig. 1, the control for Δ amyA/B/C actAp-amyB should be Δ amyA/B/C amyBp-amyB or Δ amyA/C.
- 2) Fig. 3b: The authors should include the data on bipA mRNA, which is the hacA target.
- 3) Figure 4b: It is hard to understand the decrease in the short fragment of amyA/B/C mRNA. How about quantification?
- 4) Page 4, line 24 with mutation→with a mutation
- 5) Page 6, line 21-22. The authors mentioned "After adding DTT, the mRNA level of amyB-FS was found to be not much lower than that of the intact amyB (Fig. 2c)". This sentence seems a bit confusing. For example: "Decrease in the mRNA level of amyB-FS was not observed compared to that of the intact amyB upon DTT treatment".
- 6) Discussion page 11, lines 23-25. The authors mentioned "The transcript of the frameshift mutant of amyB (amyB-FS) gradually decreased upon DTT treatment, although its rate of decrease was slower than that of the wild-type amyB mRNA". This result should be mentioned in the Results section.
- 7) Discussion, p. 12, lines 6-8. The authors mentioned "As ireA-repressed cells expressing hacAi suppressed the reduction in the amyA/B/C mRNA levels treated with DTT, we speculated that RIDD and RESS might be regulated by different pathways." This result should be mentioned in the Results section.
- 8) Discussion, page 12, line 25 - page 13, line 1. The authors mentioned "Dysfunction of RIDD due to loss of the Ski complex caused severe growth inhibition in maltose medium." They did not show the data which indicate that Dysfunction of RIDD caused severe growth.

Reviewer #1 (Remarks to the Author):

In the manuscript by Tanaka et al the authors investigate the mechanism underlying the phenomenon of reduced transcript levels of genes encoding secreted proteins under ER stress conditions in Aspergillus oryzae. Using the constitutive actA promoter, the authors demonstrate that reduced transcript levels of amyA/B/C and glaA after induced ER stress are not (only) based on the previously coined term RESS (repression under secretion stress). They furthermore demonstrate convincingly the requirement of ER targeting and the presence of IreA for degradation of the amyB and malP mRNA. Reduced degradation in the Δ ski2 background further suggests that the amyA/B/C and malP mRNAs are degraded via RIDD (regulated Ire1-dependent decay) and the 3'-5' exosomal degradation pathway. Importantly, RIDD of amyA/B/C and of malP mRNAs is induced under physiological conditions. Taken together, the authors reveal that RIDD, which was previously thought to occur only in higher eukaryotes and fungi without a functional Ire1/Hac1 pathway (S. pombe and Candida glabrata), functions in Aspergillus oryzae (and likely also other Aspergilli) to reduce the folding demand in/of the ER. This paper is very well and clearly written. Intro/abstract and discussion are on point. It has not happened very often to me that I do not have any major or minor points to be addressed. Congratulations to this wonderful piece of work adding truly novel and important insight, expanding the knowledge on the unfolded protein response in fungi and in general.

We thank the reviewer for very positive comments. We are very happy to receive such honorary comments.

Reviewer #2 (Remarks to the Author):

In this manuscript, the authors insist that Regulated Ire1-dependent decay (RIDD) occurs under physiological ER stress in Aspergillus oryzae. To support this, the authors treated DTT and checked mRNA levels of amyA/B/C. The authors also used ireA, hacA, and ski2 mutant strains and examined amyA/B/C and malP expression. It might be useful for understand the ER stress response. There are some comments.

We thank the reviewer for helpful comments. We have addressed all comments.

- The authors can detect amyB in Figure 2. Then, it seems possible to check the expression of the amyA, amyB, and amyC genes.

As described in page 5, lines 17-18, the sequences of *amyA*, *amyB*, and *amyC* are almost identical, so these genes cannot be individually detected. In Figure 2, we used an *A. oryzae* strain lacking all three copies of the intrinsic α -amylase genes ($\Delta amyA/B/C$) as a host for expression of wild type or mutant *amyB* (*amyB-FS*) genes to specifically detect these *amyB* transcripts.

- It's much better to focus on the amyB expression.

The $\Delta amyA/B/C$ strain was used only for expression of *amyB* driven by *actA* promoter (Fig. 1, right panel) or *amyB-FS* (Fig. 2C). Therefore, it is not possible to detect native *amyB* mRNA as distinct from *amyA* or *amyC* mRNAs.

- The amyB protein production or enzyme activity should be examined.

When *A. oryzae* cells are treated with DTT, the newly synthesized secreted proteins do not fold properly and may be degraded by ER-associated degradation. In addition, pre-existing ones misfold due to the reduction of disulfide bonds. These may lead to difficulties in detecting the α -amylase production or its enzyme activity.

- The cleavage pattern of amyA/B/C in Figures 1 and 3 can be provided.

As described in page 7, lines 18-20, the short fragments of the *amyA/B/C* mRNAs resulting from cleavage by IreA are rapidly degraded by the 3' to 5' mRNA degradation pathway owing to lack of a poly(A) tail. Therefore, they can only be clearly detected in a disruption mutant strain of *Aoski2*

involving in the 3' to 5' mRNA degradation pathway. Since *A. oryzae* strains used for experiments in Figure 1, 3A, and 3B are not deficient in *Aoski2*, the cleavage products of *amyA/B/C* mRNA are not detectable.

- The summary figure can be useful for understand this manuscript.

In accordance with this reviewers' comment, we show the summary figure as new Fig. 6.

Fig. 6. Schematic diagram showing the response to physiological ER stress induced by amylytic enzyme production. (a) Expression of amylytic genes is induced by maltose uptake through MalP, and the unfolded proteins resulting from the synthesis of nascent polypeptides from their mRNAs induce ER stress. (b) The resulting ER stress activates IreA, leading to HacA activation and RIDD. Thereafter, RIDD and the HacA-mediated UPR and RESS alleviate ER stress. In addition to RIDD and RESS, carbon catabolite repression (CCR) and MalP endocytosis induced by glucose generated from the maltose hydrolysis also contribute to the decrease in *amyA/B/C* and *malP* mRNAs. The correctly folded protein is represented by the three-dimensional structural image of α -amylase (PDB ID: 6TAA). The red arrows indicate the activation of transcription factors and CCR.

Reviewer #3 (Remarks to the Author):

In this manuscript, the authors found that the mechanism of regulatory Ire1-dependent decay (RIDD) exists in filamentous fungi, Aspergillus oryzae, and they also found the physiological conditions under which RIDD occurs. The authors found that RIDD of malP mRNA is strongly induced by the production of amylolytic enzymes, rather than by chemical treatments that induce an endoplasmic reticulum stress response. These results are very interesting because they show that, in Aspergillus oryzae, both UPR and RIDD are induced by physiological endoplasmic reticulum stress. Although the experiments are carefully conducted and the paper is carefully written, some parts are a bit difficult to read. The following points should be further addressed or modified.

We thank the reviewer for helpful comments. We have addressed all comments.

Major points

Fig. 3: The authors should compare the ireA mutant strain with the hacA mutant strain.

In our previous study, we failed to obtain a homokaryotic *A. oryzae hacA* disruption mutant (Tanaka et al., 2015, *Fungal Genet. Biol.*, 85, 1-6). In addition, although other research group has succeeded in obtaining *A. oryzae hacA* deletion strain, they reported that this *hacA* deletion strain shows severe growth defect and significantly reduced *amyB* mRNA level (Zhou et al., 2016, *Gene*, 593, 143-153). For these reasons, it is difficult to analyze the α -amylase mRNA in *hacA* disruption strain.

Fig. 2 and Fig. 4: The frame shift mutant applied to amyB should be analyzed with malP in Fig. 4.

The aim of the experiments in Fig. 4 is to determine whether α -amylase and *malP* mRNAs are targets of RIDD under physiological condition without DTT addition. This point is clarified by the appearance of the short fragments of *amyA/B/C* and *malP* mRNAs in the $\Delta Aoski2$ strain. If *amyB-FS* is expressed in the $\Delta amyA/B/C \Delta Aoski2$ strain, no short fragments are expected to be detected, indicating that the appearance of short fragments is the result of cleavage by IreA. However, this has already been shown by the result that the appearance of short *amyA/B/C* mRNA fragments is repressed by *ireA* repression (Fig. 4B).

Minor points

1) In Fig. 1, the control for $\Delta amyA/B/C actAp-amyB$ should be $\Delta amyA/B/C amyBp-amyB$ or $\Delta amyA/C$.

In the wild type strain, the amount of *amyA/B/C* mRNA was dramatically reduced after the addition of DTT. This indicates that mRNAs of all three α -amylase genes, including *amyB*, were reduced by the addition of DTT. Although two data (WT and $\Delta amyA/B/C actAp-amyB$) are presented side by side in Figure 1, these are independent experiments and used to elucidate that the reduction of α -amylase mRNAs upon DTT treatment is independent of the transcription of the genes. In addition, the DTT-induced reduction of *amyB* mRNA expressed from the original promoter is shown in Fig. 2C.

2) Fig. 3b: The authors should include the data on *bipA* mRNA, which is the *hacA* target.

In accordance with this reviewers' comment, we examined the *bipA* mRNA level. The *bipA* mRNA was clearly reduced in the *nmtAp-ireA* strain compared to the wild-type strain, indicating that UPR is inhibited by repression of *ireA*. Interestingly, *bipA* mRNA level was also reduced in WT+*hacAⁱ* strain. We speculate that the introduction of *hacAⁱ* has led to the abundance of chaperone proteins including BipA, thereby alleviating physiological ER stress and repressing *bipA* expression by an unknown feedback mechanism. This point needs to be investigated as a distinct study in the future. Importantly, no correlation was found between *bipA* expression level and *amyA/B/C* mRNA level, suggesting that the downstream events of HacA activation is not directly related to the reduction in *amyA/B/C* mRNA.

Northern blot analysis of *bipA* mRNA

3) Figure 4b: It is hard to understand the decrease in the short fragment of *amyA/B/C* mRNA. How about quantification?

We quantified signal intensities of *amyA/B/C* and *malP* mRNAs at 60 minutes after maltose addition and calculated the relative amounts of short mRNA fragments to full-length mRNA. The relative amounts of *amyA/B/C* and *malP* short mRNA fragments were reduced by 50% and 72%, respectively, by repressing *ireA*. This data has been shown as new Supplementary Fig. 2.

Supplementary Fig. 2. The relative amounts of short mRNA fragments to full-length mRNA. Signal intensities of *amyA/B/C* and *malP* mRNAs at 60 minutes following maltose addition were quantified using Image J software. The relative amounts of short mRNA fragments to full-length mRNA are indicated by dark bars. The error bars indicate the standard errors of three independent experiments.

4) Page 4, line 24 with mutation → with a mutation

In accordance with this comment, "with mutation" has been corrected to "with a mutation". Relatedly, "genes" on page 4, line 24, has been corrected to "gene".

5) Page 6, line 21-22. The authors mentioned "After adding DTT, the mRNA level of *amyB*-FS was found to be not much lower than that of the intact *amyB* (Fig. 2c)". This sentence seems a bit confusing. For example: "Decrease in the mRNA level of *amyB*-FS was not observed compared to that of the intact *amyB* upon DTT treatment".

We agree with this comment. This sentence has been revised in accordance with this and next comments.

6) Discussion page 11, lines 23-25. The authors mentioned "The transcript of the frameshift mutant of *amyB* (*amyB*-FS) gradually decreased upon DTT treatment, although its rate of decrease was slower than that of the wild-type *amyB* mRNA". This result should be mentioned in the Results section.

In accordance with this and previous comments, the sentences relating to this result in the Results section and the Discussion section have been revised as follows:

(Page 6, lines 21-23)

A decrease in the mRNA level of *amyB-FS* was not observed compared to that of intact *amyB* upon DTT treatment, although the *amyB-FS* mRNA was still gradually decreased (Fig. 2c).

(Page 11, lines 21-24)

Although our findings indicated that RIDD plays a major role in the reduction of transcripts encoding secretory proteins in *A. oryzae*, the slower decrease in the *amyB-FS* mRNA levels upon DTT treatment reflects the RESS of the *amyB* mRNA.

7) Discussion, p. 12, lines 6-8. The authors mentioned "As ireA-repressed cells expressing hacAi suppressed the reduction in the amyA/B/C mRNA levels treated with DTT, we speculated that RIDD and RESS might be regulated by different pathways." This result should be mentioned in the Results section.

Since this sentence expresses speculation in response to the result, we believe that it is appropriate to include it in the Discussion section. The rationale for this speculation can be found on page 7, lines 13-16, in the Result section.

8) Discussion, page 12, line 25 - page 13, line 1. The authors mentioned "Dysfunction of RIDD due to loss of the Ski complex caused severe growth inhibition in maltose medium." They did not show the data which indicate that Dysfunction of RIDD caused severe growth.

In accordance with this comment, this sentence has been revised as follow:

(Page12, lines 23-24)

Loss of the Ski complex, which accumulates mRNA remnants from the cleavage by IreA, caused severe growth inhibition in maltose medium.

REVIEWERS' COMMENTS:

Reviewer #2 (Remarks to the Author):

The authors addressed all issues raised by reviewers.

Reviewer #3 (Remarks to the Author):

The authors have addressed my previous criticisms. Therefore, I think that this manuscript is now suitable for publication.